# Board-Gender Diversity, Family Ownership, and Dividend Announcement: Evidence from Asian Emerging Economies

**Adeel Mustafa [1,*], Abubakr Saeed [2], Muhammad Awais [1] and Shahab Aziz [3]**

[1]  Department of Business & Technology, Foundation University, Islamabad 44000, Pakistan; m.awais@fui.edu.pk
[2]  Department of Management Sciences, Comsats University, Islamabad 44000, Pakistan; abubakr.saeed@comsats.edu.pk
[3]  Business Studies Department, Bahria University, Islamabad 44000, Pakistan; saziz.buic@bahria.edu.pk
[*]  Correspondence: adeelpk.research@gmail.com; Tel.: +92-322-771-1777

**Abstract:** In eras of intense debates on the appointment of women on corporate boards, this research sheds light on the structure of board in Asian emerging economies by examining how women on board of family businesses separately and collectively affect the dividend announcement of business organizations. On the basis of the panel data of four Asian emerging economies—China, Malaysia, Pakistan, and India—for the period 2010–2018, the results from our Tobit regression showed the adverse (negative) and significant impact of women on boards and in family businesses upon dividend announcement. It is important that policymakers should not view firms with one eye. There should be a spillover on board gender diversity from international to domestic levels, and international firms should be set as an example for domestic firms for the inclusion of women on boards. It might be the best time for Asian emerging economies to take productive action for balancing the gender in boardroom settings, and to set a minimum mass of women on boards for better and more effective decision making.

**Keywords:** board gender-diversity; family ownership; dividend announcement; emerging markets

## 1. Introduction

Corporate boards are responsible for major corporate decisions (Chen et al. 2017), and the effectiveness of such decisions is dependent on the characteristics of the board. Prior literature largely focuses on how board characteristics are related to corporate productivity (Hu et al. 2010; Van Essen et al. 2012), but gives little guideline on how these characteristics effect the significant strategic decision of dividend policy, as it is a major issue that corporate boards face (McGuinness et al. 2015; Gyapong et al. 2019). In perfect markets dividend policy is irrelevant (Miller and Modigliani 1961), but in imperfect markets dividends act as tool to mitigate the imperfections of markets such as agency issues (Dhanani 2005; Jensen 1986) that arise between the insiders (insiders) and outsiders (shareholders). Prior research has studied the effect of board characteristics (board composition, independence of board) on dividend policy (Schellenger et al. 1989; Adjaoud and Ben-Amar 2010; Sharma 2011). However, recently stakeholders are putting more emphasis on gender diversity. The idea has drawn attention worldwide on the basis of the view that by adding diverse directors can increase the performance of the board (Adams and Ferreira 2009). A growing stream of research explains that diversity on corporate boards not only leads to an effective decision process but also to good governance (Gul et al. 2011; Carter et al. 2003), which helps in increasing the wealth of shareholders (Hillman et al. 2007; Nielsen and Huse 2010). Extensively,

diversity with respect to gender is viewed as a significant basis for economic benefits of organizations, and their worth is progressively acknowledged by stakeholders. On the basis of the results of prior studies (Srinidhi et al. 2011; Harjoto et al. 2014; Gul et al. 2011; Krishnan and Parsons 2008; Bernardi et al. 2006), many countries across the globe have made it mandatory for their organizations to appoint a woman for increasing the trust and worth of shareholders. According to the Deloitte Report on Women on Boards (2019) women currently hold 16.9% of board chairs internationally, which is 1.9% more in comparison to 2017. With respect to a research report of the Association of Psychological Sciences, it was found that women are double in number (34%) in countries where there is a quota for appointment of women on boards, as well as heavy penalties in cases of non-compliance in comparison to countries that do not have any kind of regulation for appointment of women (18%). In the case of developed countries, this ratio rose from 5% to 12% between 2001 and 2012.

Many studies have explored the demographic features of women directors such as age, education, work, experience, and their links to external parties (Hillman et al. 2007; Mahadeo et al. 2012). Many researchers have also studied the positive impact of female representation on corporate boards, that is, firm performance (Adams and Ferreira 2009; Carter et al. 2010), value (Ahern and Dittmar 2012; Erhardt et al. 2003), firm philanthropy (Wooldridge 2002), control function, and social accountability (Bear et al. 2010). However, some studies have proposed that representation of women on a corporate board is dependent on the specific aspects of organization. For instance, Hillman et al. (2007) suggested that size and type of organization significantly affects the appointment of women on the corporate board of an organization. Likewise, De Cabo et al. (2012) identified the firm characteristics that explain that a high ratio of women on boards face low risk of financial distress. Regardless of the statistics that firm features of board gender diversity have filled in the research gap, it has been seen that most work has been done in developed economies (Virtanen 2012; Ely and Meyerson 2000; Kurtulus and Tomaskovic-Devey 2012; Dezsö and Ross 2012), and emerging economies are still an unexplored area.

Consequently, the objective of this paper was to fill the gap by exploring the effect of female representation on boards of Asian emerging economies. Recently, in July 2017, Pakistan (an emerging economy) made an amendment in law to appoint a woman on company boards. Hence, it is timely to study the impact of women on corporate boards of Pakistani firms and other Asian economies to decide the specific quota on company boards that can affect the performance of firms. Second, it recognizes the situations in which emerging economies disallow women on their boards and how family presence on firms' boards affect the relationship between gender diversity on boards and dividend announcement of firms. Lastly, it extends the prior literature on the basis of agency and resource dependency theory, and explains the significance of adding women on corporate boards of organizations.

Using the sample of Asian developing economies' firms for the period 2010–2018, our Tobit regression outcomes showed the negative effect of board gender diversity and family presence on the dividend announcement of firms.

Academic research has reported mixed results of the impact of board gender diversity on firms. Adjaoud and Ben-Amar (2010) and Gyapong et al. (2019) explain a positive association among board structure and dividend payments. Other researchers have documented the negative effects of women on boards due to their risk-averse behavior, lower confidence, weak decisions, and less control on boards (Arano et al. 2010; Levi et al. 2011), which affect firm value. Likewise, shareholders are reluctant to invest in a firm with female on its board (Abdullah et al. 2016; Ahern and Dittmar 2012). In this context, this study makes three valuable contributions in existing literature. Firstly, this study addresses the board gender diversity matter in Asian emerging economies by concentrating on firm features that are exclusive to their business framework and how this diversity affects the dividend announcement activity of firms. Secondly, it recognizes the situations in which emerging economies' firms disallow women on their boards and how family presence on firms' boards affect the relationship between gender diversity on boards and dividend announcement of firms. Thirdly, it extends the prior literature

on the basis of agency and resource dependency theory and explains the significance of adding women on corporate boards of organizations.

The remainder of this paper is structured as follows: Section 2 presents the literature review relating to board diversity and hypothesis development, Section 3 presents the dataset and methodology, Section 4 focuses on the empirical findings, and the last section presents the discussion and conclusion.

## 2. Literature and Hypotheses Development

### 2.1. Theoretical Literature

Resource dependency theory gives some critically important facts regarding the appointment of women on boards. It explains that boards create the link between a firm and its resources. This link offers different benefits such as legitimacy to the organization, guidance, and communication links to constituents of significance of the organization (Pfeffer and Salancik 1978). It has been contended that women on boards can bring all these benefits to firms. On the basis of the theory of resource dependency, Daily et al. (1999) explained that to increase the overall legitimacy and to motivate the existing and potential female workers, organizations should appoint women on their boards. Women on boards also offer a valuable form of realism with respect to other stakeholders, that is, customer-oriented organizations appoint more females on corporate boards to attain the business objectives in terms of the beliefs of their customers (Brammer et al. 2007). Boards with gender diversity give unique sets of information such as difference of beliefs, views, and experience, which helps in making better decisions (Lyman et al. 1985; Carter et al. 2010). Another ability of women directors that is better than in men is effective communication, which helps in doing business with other entities. Hillman et al. (2007) claim that women on boards not only serve organizations better but that they also inspire other females in the organization and help them in developing their careers by providing good mentoring. Moreover, women executives can communicate with female customers in a better way, which in addition helps to create market-oriented policies (Liu et al. 2013).

Agency theory argues in terms of the relationship of agent (management) and principal (shareholders). On the basis of this argument, directors on boards work to align (support) the benefits of management and stockholders by observing the attitude of management and the quality of their decisions (Fama and Jensen 1983). In this perspective, board independence is necessary to work in the best interest of share-holders. On the basis of the demographic features of directors, Campbell and Vera (2008) proposed that directors having diverse experiences might raise queries that normally do not come from the directors of the same background. Adams and Ferreira (2009) explain that women have a tendency to pay more attention to the working of firms and are normally more interested in monitoring teams. Similarly, Gul et al. (2011) explain that women on boards increase the financial transparency and disclosure of firms. However, studies suggest that strict monitoring not only affects the working of management but also reduces the shareholders' value (Almazan and Suarez 2003). Sonnemans et al. (2000) criticized agency theory and challenged the progressive effect of gender diversity on boards. On the basis of qualitative research, Fitzsimmons (2012) argued that organizations generally appoint women to enhance governance or to touch the gender parity level. Likewise, Broome et al. (2011) conducted qualitative research on women directors and explained that organizations pursue gender diversity on boards for improved monitoring function.

### 2.2. Empirical Literature

Following the approach of Gyapong et al. (2019), we conducted a systematic literature review and provide a summary in Table 1. Dividend payout policy of firm explains the level of legal protection of shareholders (La Porta et al. 1998). In the presence of strong legal protection, management is pressurized to pay dividends. On the contrary, in countries where shareholder protection is low, dividend payment is on the will of management and controlling shareholders. Chen et al. (2017) suggested that the level of protection of minority shareholders explains the mechanism of corporate

governance and their effect on dividend policy. Academic research has reported mixed results of the impact of board gender diversity on firms. Adjaoud and Ben-Amar (2010) explained a positive association among board structure and dividend payments. Gyapong et al. (2019) also reported the same results of positive association of female directors on dividend policy. A limited amount of research has also examined the direct relationship between gender diversity on boards and dividend policy (Byoun et al. 2016). The results of this study are consistent with Chen et al. (2017), who reported a positive impact of board gender diversity on dividend policy.

**Table 1.** Summary of studies on board gender diversity and its impact on firms' financial decisions.

| Authors | Key Findings |
|---|---|
| Cox (1991) | The results indicate that board gender diversity brings cost to an organization in the shape of communication and professional conflicts, which affect the decision-making process of board negatively. |
| Richard et al. (2004) | Board gender diversity increases intra-group conflicts, which affect firms' decision-making processes negatively. |
| Rose (2007) | Results suggest that board gender diversity does not create any influence on firm performance. |
| Francoeur et al. (2008) | The results suggest that gender diversity on boards positively affect firm value. |
| McGuinness et al. (2017) | The results suggest that women on boards enhance the social performance of the firm. |
| Nasrum (2013) | Ownership structure and corporate governance have a positive influence on dividend policy and firm value. |
| van Essen et al. (2015) | Diversification, internationalization, and financing strategies positively mediate the performance of family firms, but after the first generation, performance of family firms drops due to conventional patterns of strategic decision-making. |
| Haldar and Raithatha (2017) | The results of the study explain that the board of directors, including executive and non-executive directors, influence financial strategies of firms positively, and that audit committee plays a crucial role in increasing the effectiveness of decisions of firms. |
| Chen et al. (2017) | The results suggest that female directors pay high dividends in firms, whereas the governance system is weak and is used as a governance device. |
| McGuinness et al. (2015) | The results of the study show that independent women directors have limited influence on dividend payout, but that this impact is high and positive in firms where state investment is high. |
| Campbell and Vera (2010) | The results of the study suggest that announcement of appointment of women on boards increases the value of the firm in the long run, and that the stock market reacts positively on the appointment of women in the short run. |
| Campbell and Vera (2007) | The results of the study suggest that the relationship between women on boards and firm value is insignificant. |
| Al-dhamari et al. (2016) | The results of the study suggest that women on boards positively influence the dividend policy of firms, but only when firms have a high percentage of free cash flow. |
| Adjaoud and Ben-Amar (2010) | The results of the study suggest that firms with strong corporate governance practice positively influence the firm dividend payout policy, but that there is a negative relationship between the dividend policy and the level of firm risk. |
| Al-Rahahleh (2017) | The results of the study suggest that board gender diversity has a positive impact on the dividend policy of a firm. |
| Gyapong et al. (2019) | The results of the study suggest that female directors on boards positively influence the dividend policy of firms. This association of women and dividend policy is strong where the percentage of women on boards is high, and that the association is negative when ownership concentration is high. |
| Pucheta-Martínez and Bel-Oms (2015) | The findings of the research suggest that percentage of women and shares held by women are positively associated with dividend policy, but that institutional female directors have a negative impact on dividend policy, whereas independent and executive women have no effect on the dividend policies of firms. |
| Djan et al. (2017) | The results suggest that capital structure as an interactive term has insignificant impact on dividend policy. |
| Elmagrhi et al. (2017) | The findings of the study suggest that board gender diversity has a negative impact on the dividend policy of the firm, whereas audit committee and board size have a positive impact on the dividend policy of a firm. |
| Sanan (2019) | The results of the study suggest that percentage of female directors on boards have a negative effect on dividend policy, and that firms with strong corporate governance practices pay low dividends. |
| Ye et al. (2019) | The results of the study show that board gender diversity facilitates the corporate governance practice and promotes dividend policy. Institutional environments weaken the role of gender diversity on dividend policy, and relationship between gender diversity and dividend policy is strong when females have ownership in a firm. |
| Attig et al. (2016) | The findings suggests that family ownership has a negative impact on the dividend policies of firms. |
| Benjamin and Biswas (2017) | The results suggest that board gender diversity has a positive impact on the dividend policies of firms, but that the positive relationship only exists in the firms without CEO duality. |
| Setia-Atmaja (2010) | The results suggest that high investor protection pressurizes controlling shareholders to pay more dividends. |

Other researchers have documented the negative effects of women on boards due to their risk-averse behavior compared to their male counterparts (Arano et al. 2010). Firms with women on their boards face more problems of underinvestment (Levi et al. 2011), which affect shareholder wealth in the long run and increase agency issues. Likewise, shareholders are reluctant to invest in firms with females on boards due to their lack of confidence, power, and influence (Abdullah et al. 2016).



Existing evidence provides the indication that firms with women-dominant corporate boards show low performance and value (Ahern and Dittmar 2012).

### 2.3. Hypotheses Development

#### 2.3.1. Women Directors on Boards

Women on the corporate board of firms can deliver a diverse perspective and ideas to board decisions about policies (Hillman et al. 2007). Gender diversity on boards is considered significant in terms of protecting the benefits of all stakeholders and can be used to improve the financial policies of firms, improving the board functioning and efficiency due to diverse skills, including knowledge and other qualities (Carter et al. 2003). Women directors normally engage in risk-averse behavior (Harjoto et al. 2014), and few are overconfident in comparison to male counterparts on boards (Dowling and Aribi 2013). Adams and Funk (2012) suggest that risk-averse behavior of women can vanish and their role on boards can be reduced significantly if they accept the dominancy of males on boards. Women on boards improve effectiveness and efficiency of firms due to their hardworking nature and skills (Ittonen et al. 2010). These skills and abilities help women to take the actions that are necessary for reducing agency issues in firms. The literature suggests that women use dividend payments to reduce agency issues and have a strong positive impact in financial strategies of companies such as in dividend payment decisions. The high proportion of women on boards results in better financial strategies (Harjoto et al. 2014). They provide exclusive ideas about complex issues (Virtanen 2012), and show professionalism and integrity in their dealings (Ingley and Walt 2005), which displays their dynamic leadership style (Eagly and Johnson 1990). Female directors can observe the behavior of directors well by discussion, communication, and attendance (Srinidhi et al. 2011), which are dire to investigate the opportunistic behavior of managers (Adams and Ferreira 2009). According to Fondas and Sassalos (2000), women directors believe that due to their presence on boards, stakeholders expect much from them, and that it is the duty of female directors to work in the best interest of shareholders. Gul et al. (2011) explains that women are much better in reducing the asymmetric information and in maintaining more transparency in firms than males. Earning management is much higher in companies where their board is gender-diverse (Krishnan and Parsons 2008). At the board level, existence of women can strongly affect board decisions (Hoogendoorn et al. 2013). Gender diversity on boards not only brings in ethical values but also (Kramer and Konrad 2007) novel ideas to firms (Bernardi et al. 2006), and compacts the influence of male directors, making board more answerable for their actions (Terjesen et al. 2009). However, in the civil law environment, women executive directors have a strong benefit in supporting the management for their benefits on the cost of shareholders due to a weak monitoring mechanism (Prasanna 2014). Women executive directors could play a role in the reduction of dividend payment because they do not need it as a monitoring device (Prasanna 2014; Van Pelt 2013; Francis et al. 2014). Earlier studies (Hsu and Wu 2014; Cho and Kim 2007; Cheng and Courtenay 2006; Ruiz-Barbadillo et al. 2007; Zhang 2008; Deshmukh et al. 2010; Banerjee et al. 2013) explain the negative effect of executive directors on dividend announcement because executive directors increase the supremacy of management on boards and support low-dividend payout decisions. The literature suggests that the nature of women is not different from male directors on boards (Petrides and Furnham 2006; Baack et al. 1993; Hardies et al. 2014) in terms of management style (Powell 1990), behavior (Morrison et al. 1987), and in always feeling the need for more planning. Thus, according to the above arguments suggested by the literature (Francis et al. 2014; Beneish 2001), we can posit that women directors on boards negatively affect the dividend announcement of firms.

**Hypothesis 1.** *Women on boards negatively affect the dividend announcement.*

2.3.2. Family Ownership and Board Gender-Diversity

In emerging economies, there is a solid faith that women can play an important role in reducing the problems in family business (Evans 2005). Women's efforts in business and family are not well recognized due to their defined traditional role (Lyman et al. 1985). Many researchers explain that women in higher positions in family businesses are hardly considered serious (Dumas 1998). Investigating the basic causes, Clark (2000) explains that women in traditional families face some serious barriers from parents and siblings especially when they move to perform senior roles in family firms. On the basis of this argument, studies have explained that fathers create more obstacles for daughters, especially when they are in positions to take over family businesses. Therefore, daughters are normally ignored in terms of performing the work of top-level positions and are pressurized to perform a supporting role for their brothers. In particular, preference for male candidates is high in cases of Asian developing economies (Deng 2016). Ramachandran and Bhatnagar (2012) explain that domination of males in family businesses is normal, and that the firstborn son considers it his right to take over the business from his father. The significance of social values is obvious in managerial practices in Asian emerging economies. This contains centralized decision processes, protective leadership styles, respect for the oldest peers and their desires, and solid family supervision (Prasad et al. 2013). Likewise, family businesses in China are highly dependent on social values, claiming consistency on the role of gender (Yan and Sorenson 2004). The system of social and ethical philosophy explains that interest of family in family businesses should be the first priority over individual interest, and that the social pyramid should be followed. This indicates that a woman in a family should not focus on her own career but work for the interest of the family and provide support to male members of the family. Likewise, Indian family businesses are also male-dominant, and the eldest member of the family is treated as the manager (termed as *Karta*), and the male member is expected to hold the highest positions in terms of management, whereas the role of women is restricted only to working as a helper (Ramachandran and Bhatnagar 2012). Overall, family companies in Asian emerging markets are male-dominant, and the duty of females is to only work as an organizer or supporter. Earlier studies suggest that women can act more than a rubber stamp (McGuinness et al. 2015) on family firms. The active role of women on boards gives the sense of protection to shareholders (McGuinness et al. 2015) and has a better tendency in terms of aligning the economic benefits of managers and shareholders. If the role of women is effective on boards, they can affect the financial decisions of the firm, specifically, the dividend policy (Adams and Ferreira 2009; Chen et al. 2017). The effect of family ownership on the relation of board gender diversity and dividend policy can be explained by two perspective factors: interest alignment and exportation of minority shareholder resources for personal benefit (Gyapong et al. 2019). In the interest alignment perspective factor, the main interest of all shareholders is the maximization of return on their investment (Su et al. 2013). Accordingly, controlling shareholders aligns their interest with external shareholders. The reason behind this could be the strong implementation of the law that prevents them from misusing the resources of minority shareholders. In this situation, they are encouraged to perform activities that ensure the benefits of all shareholders. Thus, controlling shareholders becomes the bridge between management and shareholders to reduce the agency issue (Konijn et al. 2011; Su et al. 2013). In the exportation perspective factors, family owners can exploit the resources of minority shareholders for their personal benefit by reducing the dividend payments due to their high ownership concentration and active participation in company matters (Shleifer and Vishny 1986; Bhojraj and Sengupta 2003; Renneboog and Szilagyi 2015). The contribution of family firms in the economy is significantly high, and globally more than 50% of firms are controlled by family groups using cross-stock holding in different firms (La Porta et al. 1998) where they also affect the financial policies of firms such as dividend policies (Ramli 2010). Family businesses are much more concerned about the ongoing concerns about the nature of business (Prencipe et al. 2011), creating agency issues by working in their own interest. The literature suggests that family businesses use dividend announcement to satisfy shareholders who have a minimum stake in a firm and use their resources for their own benefit and ignore any kind of

investment opportunities to reduce the risk of weakening control (Koropp et al. 2014; Mahérault 2000). To control and use the firm for their own benefit, top managers are normally selected from family (Faccio et al. 2001; Yoshikawa and Rasheed 2010). Due to their controlling role in firms, they strongly influence the value of firms, creating the agency issue (Blanco-Mazagatos et al. 2007). However, professionalism in management and positive use of human capital in family firms can positively affect its policies, which can be attained by creating gender diversity on boards. On the basis of the above arguments, we can posit that family ownership in Asian emerging economies has a negative (adverse) impact on corporate board gender diversity and dividend announcement.

**Hypothesis 2.** *Family ownership in Asian emerging economies negatively moderates the relationship between corporate board gender-diversity and dividend announcement.*

## 3. Data and Methods

### 3.1. Context Overview

To test our suggested hypothesis, we selected firms from four Asian emerging economies: China, Pakistan, India, and Malaysia, on the basis of the emerging economy index of Morgan Stanley Capital International (MSCI). MSCI is an investment research firm that offers indices to measure equity market performance internationally. The selection of Asian emerging economies was based on several reasons. Firstly, rapid growth in recent times makes them a best representative of Asian emerging economies. According to the World Trade Organization (2019), these countries are significant players in international markets due to their human resource and rapid growth. The literature suggests that the progress of these countries will affect the growth of developed economies negatively, and that they will represent the world economy by 2030 (Wilson and Purushothaman 2003; Saeed and Sameer 2017). Secondly, to increase the investor confidence in firms, these countries have taken a progressive measure in the field of reporting and corporate governance. Lastly, India, China, and Pakistan are basically male-dominant societies, and female representation on corporate boards is at negligible level (e.g., agriculture, forestry and fishing, construction, manufacturing, mining, transport, utilities), whereas in some industries women are at the highest level (e.g., fashion, human resources, financial institutions). In male-dominant societies, women are discriminated against and lack free movement from domestic to public life (Lee 2015). Due to controlled and self-established traditions, mobility of women is completely banned in most parts of India and Pakistan. On other hand, in the case of China, narrow space has been given to women in routine life, with the main objective being to serve family. Due to the strict policy of one child families, boys are given more value in comparison to girls, as girls are considered a burden. On the other hand, in the case of Malaysia, women are rapidly increasing in career professions, and this rise has been reported in experienced women professionals across industries (Ismail and Ibrahim 2008), with one of the main reason for this being education. Gender inequality in education leads major challenges towards employment and economic growth of a country (Klasen and Lamanna 2009). According to the World Economic Forum (2018), Malaysia has attained the desired education level in comparison to 2006, whereas in the case of the gender gap and economic participation of women in the economy, the country is still working towards this and making positive changes. According to the report, India and Pakistan are at the lowest scale with respect to these three variables of female participation, whereas China is working to give better opportunities to women. Different countries have passed laws for decreasing the gender gaps, such as recently in 2017 where Pakistan made an amendment in law to appoint at least one woman on the corporate board of firms in trying to give equal employment opportunities to females. Currently, women in Pakistan hold 9.07% of board seats, which is the second-lowest in comparison to Korea where female representation is only 4.1%, whereas Malaysia stands in the highest place with a total of 13.9% of female representation on corporate boards (Deloitte Report on Women on Board 2019).

### 3.2. Sample

The data for the study was gathered from OSIRIS database provided by Bureau van Dijk (BvD), a global specialist of corporate information, containing information on over 38,000 listed companies, banks and insurance companies around the world. Information regarding boards was primary hand-collected from the annual reports available on OSIRIS under the head of the board of directors, whereas in some cases it was gathered from companies' websites. The companies that have missing information related to analysis were excluded. Data were collected from 2010 to 2018. Furthermore, the firms that have three consecutive years of data were included in a sample of the overall sample contains unbalanced panel observation from 2000 listed firms—500 from each of the countries of Pakistan, India, Malaysia, and China. The gender of directors was measured by their first name and, to lessen slippages, and we used numerous name thesauruses, comprising some linguistic-specific forms (Hindi, Chinese, English, Urdu, and Malay). We also consulted proxy statements and companies' annual reports where director names were abbreviated. We also spotted the titles, such as he, she, Miss, Mrs., and other measures to determine the gender of the directors.

### 3.3. Variable Measurement

#### 3.3.1. Dependent Variable

Our dependent variable was the dividend announcement, as it is the most important tool that not only reduces the agency problem (Yarram and Dollery 2015) but also helps in creating the economic and financial value of the firm. Dividend announcement reduces the free cash available to managers (Ben-Nasr 2015; Firth et al. 2016), and it also has been observed that firms with a low level of transparency pay high dividends to satisfy their shareholders (Bhattacharya 1979).

#### 3.3.2. Independent Variables

We used two explanatory variables, in line of previous literature regarding board gender diversity (Adams and Ferreira 2009; Nielsen and Huse 2010; Boulouta 2013). Percentage of women in the corporate board of firms was used as our first independent variable, which was calculated as females on board divided by board size. Our second variable was family ownership, which was the dummy variable. The value of 1 was assigned if the firm was family-owned, otherwise it was assigned 0. The literature on family firms is so wide, and thus it is difficult to explore the exact definition of family firms in which all stakeholders agree. However, if a family holds 10% or more shares in company (Anderson and Reeb 2003; Al-Najjar and Kilincarslan 2016), participates in daily management of firms, and holds more seats on the board, then firm is recognized as a family firm (Miller et al. 2007). Therefore, on the basis of this explanation, we recognized the firms as family firms if founders and their families held more seats on the board, and this was done using the annual reports and internet information of firms.

#### 3.3.3. Control Variables

Moreover, to test our hypothesis we used some controlled variables that were generally used in gender studies (Nielsen and Huse 2010; Hillman et al. 2007). Firm size is defined as the logarithm of total assets. Return on assets is the profit before taxes by total assets. The financial leverage ratio was calculated using total debt divided by shareholders' equity. Board dependence was calculated using the percentage of independent women directors divided by board size. Percentage of women on the board was calculated using the number of women on the board divided by the board size. Directors were considered executives if they were fully dependent on the firm, and worked as a full-time employee of the company, enjoying all benefits that employees received. On the other hand, independent directors were considered as those who did not have any concern with the firm and were fully independent in their decisions Board size consisted of the total number of directors including male and female holding

chairs on the board. Liquidity ratio was used to measure the firm ability to pay its short-term debts. To address the industry and time variations, we used the dummy variables.

*3.4. Method of Analysis*

Our key dependent variable was the dividend announcement of the firm. A value of 1 was assigned to companies where firms announced dividend, and 0 if not, and was continuously scattered over positive values. Therefore, there was a possibility that the ordinary least squares estimator generated negative values. Furthermore, a dependent variable that showed the value of 0 did not have a restricted normal distribution, which implied that our interpretations would have only asymptotic explanations. Hence, the Tobit regression model was more apt for estimation in our research, as it allowed better specific distribution of our dependent variable (Wooldridge 2002). Normally this model is used where data is censored. According to Jizi et al. (2014), if the values of the dependent variable were grouped in some observations at a limiting amount, which is normally zero, the Tobit model becomes more powerful in comparison to other regressions models as it makes use of all observations, irrespective of whether they are at the limit or above. Previous studies have explained the concerns about omitted variables that are raised due to joint determination of female presence on boards and firm features. It is possible that both variables affect women directors. Furthermore, there was a possibility that board gender diversity may have been affected by un-observed firm-level variables such as corporate environment and managerial preference, which lead to the problem of omitted variables biases; therefore, obtaining unbiased results is a real challenge.

To examine the relation among dividend announcement and board gender diversity, we applied the Tobit regression model. Prior studies on dividend payout policy have used a similar estimation technique (Attig et al. 2016; Saeed and Sameer 2017; Al-Najjar and Kilincarslan 2016; Lam et al. 2012). The model specification to test our hypotheses was as follows:

$$\begin{aligned} Dividend_{it} = \ & \alpha + Board\ Gender\ Diversity_{it} + \beta_2\ Family\ Ownership_{it} + \\ & \beta_3\ Board\ Gender\ Diversity_{it} \times Family\ Ownership_{it} + \beta_4\ Board\ independence_{it} + \\ & \beta_5\ Executive\ women_{it} + \beta_6\ Return\ on\ assets_{it} + \beta_7\ Financial\ leverage_{it} + \beta_8\ Liquidity_{it} \\ & + \beta_9\ Firm\ size_{it} + \beta_{10}\ Board\ size_{it} + CountryDummies + YearDummies_{it} + \varepsilon_{it}\ldots\ldots\ldots \end{aligned} \quad (1)$$

where dividend is the dependent variable. CountryDummies represents country fixed effects, YearDummies represents time fixed effects, and $\varepsilon$ represents the error term.

## 4. Empirical Findings

This section discusses the descriptive stats, correlation matrix, and regression estimation results of models, which assess the impact of women on boards and family presence on the dividend announcement of firms.

Table 2 presents the percentage of women directors across Asian emerging economies on the basis of our sample. It can be seen that 49% of Pakistani and 33.4% of Chinese companies did not have women on their boards; this percentage was quite high in comparison to other Asian emerging economies. However, in the case of Indian and Malaysian companies, the percentage of no women on boards was 10% and 15.4%, respectively, of the entire sample, which shows that Indian and Malaysian authorities are taking strong measures for the inclusion of women on the corporate board of organizations.

**Table 2.** Presence of female directors in Asian emerging economies (in the years 2017–2018).

| No. of Females on Firms Board | China | Malaysia | India | Pakistan |
|---|---|---|---|---|
| Composition of Board | Frequency: No. of Companies (%) | | | |
| Entire sample | | | | |
| 0 female directors | 167 (33.4%) | 77 (15.4%) | 50 (10%) | 245 (49%) |
| 1 female director | 141 (28.2%) | 126 (25.2%) | 167 (33.4%) | 132 (26.4%) |
| 2 female directors | 77 (15.4%) | 105 (21%) | 133 (26.6%) | 63 (12.6%) |
| 3 female directors | 40 (8%) | 6 (15.2%) | 83 (16.6%) | 39 (7.8%) |
| More than 3 female directors | 75 (15%) | 116 (23.2%) | 67 (13.4%) | 21 (4.2%) |
| Total | 500 (100%) | 500 (100%) | 500 (100%) | 500 (100%) |

Further, 26.4% of Pakistani companies had one female director, 12.6% had two, 7.8% had three, and only 4.2% of firms had three or more than three women on their boards. In the case of Malaysia, 25.2% of firms had one woman on their board, 21% had two, 15.2% had three, and 23.2% had three or more women on their boards. The percentage of three or more women on boards in the case of Malaysia was high in comparison to the other Asian emerging economies of India and China, which were 13.4% and 15%, respectively, but lowest in the case of Pakistan with only 4.2%. Earlier studies suggested that the mass of women on boards plays a critical role on boards (Torchia et al. 2011), and that one woman is considered as a token (Elstad and Ladegard 2012). Furthermore, presentation of women in family firms of Asian emerging economies is presented in Table 3. In the China sample, 21% of family firms had no women directors on their boards, 35% had one woman, 19% had two, 5% had three, and 19% had more than three women on their board. In the case of Malaysia, 12.7% of firms of the entire sample had no women on their boards, 24% had one woman, 22.1% had two, 15.9% had three, and 25.2% had more than three women on their boards. In the case of India, 2.6% of firms had no women on their boards, 36.3% had one woman, 25.8% had two, 20.3% had three, and 15% had more than three women on their boards. In the case of Pakistan, family firms with no women on their boards were 48.7% of the entire sample, which was quite high compared to other Asian emerging economies, with 31.6% having one woman on their boards, 10.3% having two females on their board, 6.9% having three women, and 2.5% having more than three women on their board, which was the lowest percentage compared to the boards of other countries, suggesting that getting a board position is quite difficult for women in firms, and is tougher in cases of family-controlled firms. Table 4 provides summary statistics and correlation coefficients of Asian emerging economies.

**Table 3.** Presence of female directors on family firms of Asian emerging economies (in the years 2017–2018).

| No. of Females on Family Firms Board | China | Malaysia | India | Pakistan |
|---|---|---|---|---|
| Composition of Board Entire Sample | Frequency: No. of Companies (%) | | | |
| 0 female directors | 25 (21%) | 45 (12.7%) | 6 (2.6%) | 57 (48.7%) |
| 1 female director | 42 (35%) | 85 (24%) | 84 (36.3%) | 37 (31.6%) |
| 2 female director | 23 (19%) | 78 (22.1%) | 60 (25.8%) | 12 (10.3%) |
| 3 female directors | 6 (5%) | 56 (15.9%) | 47 (20.3%) | 8 (6.9%) |
| More than 3 female directors | 23 (19%) | 89 (25.2%) | 34 (15%) | 3 (2.5%) |
| Total | 119 (100%) | 353 (100%) | 231 (100%) | 117 (100%) |

**Table 4.** Values of variables.

| Variables | Mean | SD | 1 | 2 | 3 | 4 | 5 | 6 | 7 | 8 | 9 | 10 |
|---|---|---|---|---|---|---|---|---|---|---|---|---|
| Dividend announcement | 0.69 | 0.47 | - | | | | | | | | | |
| Female presence | 0.77 | 0.41 | 0.04 *** | - | | | | | | | | |
| Family presence | 0.43 | 0.49 | −0.02 * | −0.04 *** | - | | | | | | | |
| Financial leverage | 89.96 | 134.54 | −0.07 *** | 0.01 * | 0.04 *** | - | | | | | | |
| Return on assets | 6.78 | 9.93 | 0.29 *** | 0.00 | −0.04 *** | −0.20 *** | - | | | | | |
| Liquidity | 1.21 | 1.63 | 0.053 *** | −0.00 | −0.03 *** | −0.18 *** | 0.17 *** | - | | | | |
| Firm size | 6.31 | 2.12 | 0.18 *** | 0.20 *** | 0.14 *** | 0.19 *** | −0.04 *** | −0.06 *** | - | | | |
| Board size | 2.89 | 0.54 | 0.06 *** | 0.18 *** | −0.08 *** | 0.05 *** | 0.04 *** | −0.04 *** | 0.21 *** | - | | |
| % independent women on board | 1.51 | 3.12 | 0.06 *** | 0.22 *** | −0.15 *** | −0.03 ** | 0.06 *** | 0.06 *** | 0.01 | 0.06 *** | - | |
| % executive women on board | 9.20 | 8.96 | 0.01 | 0.47 *** | −0.06 *** | −0.04 *** | 0.00 | 0.05 *** | −0.05 *** | −0.16 *** | 0.332 *** | 1 |

\* Denotes $p < 0.05$, \*\* denotes $p < 0.01$, \*\*\* denotes $p < 0.001$.

With respect to the correlation between variables, there was a negative relationship between dividend announcement and women on boards in the presence of family firms. There was no strong correlation between the firms, which showed that there was not a serious problem of multicollinearity in our regression. In Table 5, we compared the means of the firm having or not having a female presentation on their boards. By presence of female on boards, we refer to the having of at least one woman on a board.

**Table 5.** Comparison of firms with and without female directors in Asian emerging economies.

| Variables | Mean for Firm without Female Directors | Mean for Firm with Female Directors | t-Stats |
|---|---|---|---|
| 1. Dividend announcement | 0.47 | 0.638 | −0.167 * |
| 2. Family presence | 0.48 | 0.421 | 0.062 * |
| 3. % women on board | 1.04 | 11.18 | −10.14 * |
| 4. % independent women on board | 0.17 | 1.90 | −1.72 * |
| 5. Board size | 2.69 | 3.01 | −0.31 * |
| 6. Firm size | 5.54 | 6.80 | −1.26 |
| 7. Return on assets | 6.41 | 6.43 | −0.02 * |
| 8. Liquidity | 1.20 | 1.19 | 0.01 * |
| 9. Financial leverage | 79.27 | 88.70 | −9.42 |

\* Denotes $p < 0.001$.

We have seen through these comparisons that firms having female representation on their board are much older and larger compared with the other firms in our entire sample. These characteristics suggest that having females on corporate boards could be shaped by firm features. The Tobit regression results for Asian emerging economies are presented in Table 6 collectively. We ran three models separately to see the impact of women on boards on the dividend announcement of the firm. Model 1 solely deals with the female presence and its effect on dividend announcement, model 2 presents the effect of female presence and family presence separately, whereas model 3 contains our main effect model with control variable.

**Table 6.** Regression results (dependent variable: dividend announcement) for Asian emerging economies.

| Variables | Model 1 | Model 2 | Model 3 |
|---|---|---|---|
| Female presence | −0.0714 * | −0.0724 * | −0.0251 |
| | (0.0396) | (0.0396) | (0.0445) |
| Family presence | | −0.0540 * | 0.0190 |
| | | (0.0296) | (0.0432) |
| Female presence × family presence | | | −0.108 ** |
| | | | (0.0465) |
| Financial leverage | −0.000490 *** | −0.000489 *** | −0.000488 *** |
| | (0.000103) | (0.000103) | (0.000103) |
| Return on assets | 0.0518 *** | 0.0519 *** | 0.0520 *** |
| | (0.00180) | (0.00181) | (0.00181) |
| Liquidity | 0.000134 | 0.000149 | 0.000182 |
| | (0.00824) | (0.00823) | (0.00824) |
| Firm size | 0.150 *** | 0.150 *** | 0.150 *** |
| | (0.0103) | (0.0103) | (0.0103) |
| Board size | −0.000650 | −0.000603 | −0.000527 |
| | (0.00142) | (0.00142) | (0.00142) |
| % of independent women on board | 0.00894 * | 0.00895 * | 0.00900 * |
| | (0.00504) | (0.00504) | (0.00505) |
| % of executive women on board | 0.00382 ** | 0.00383 ** | 0.00394 ** |
| | (0.00192) | (0.00192) | (0.00192) |
| Constant | −0.872 *** | −0.835 *** | −0.863 *** |
| | (0.0910) | (0.0934) | (0.0942) |
| Observations | 10,707 | 10,707 | 10,707 |
| Likelihood ratio chi$^2$ | 1677.61 | 1680.93 | 1686.33 |
| Probability > chi$^2$ | 0.0000 | 0.0000 | 0.0000 |
| Pseudo $R^2$ | 0.1266 | 0.1269 | 0.1273 |

*** Denotes $p < 0.01$, ** denotes $p < 0.05$, * denotes $p < 0.1$.

Our first hypothesis suggested that women had a negative effect on the dividend announcement of the firm. In the case of Asian emerging economies, the coefficient of board gender diversity was negative and significant, and results for Asian emerging economies collectively supported our first hypothesis. Hypothesis 2 posited a negative relationship between board gender diversity and dividend announcement in the presence of interactive term, which was family ownership. In our second model, the results showed the negative coefficient value, and even when we included the interactive term the coefficient was negative and significant, which supported our hypothesis. Thus, results for Asian emerging economies strongly support hypothesis 2 collectively. However, when we looked at the coefficient of interactive terms in the model, it suggested that negative effects of family ownership can be reduced by operating the firms in international markets. The coefficient of interactive terms was positive for all Asian emerging economies, which supported our hypothesis. Similarly, a negative effect of board gender diversity on dividend announcement could also be reduced by international operations of firms.

We treated the women on the corporate boards as a homogenous group in our research, although prior studies have explained that it is easy for women to get an independent membership on boards in comparison to executive seats (Kellerman and Rhode 2007).This clearly explains that getting a top position on a board in the presence of male counterparts is not easy for women. The data sample also endorsed this opinion. On this basis, our sample women held only 7.9% seats on corporate boards of firms, where Malaysia had the highest and India and Pakistan had the lowest representation.

## 5. Discussion and Conclusions

Currently, it is widely believed that the presence of women on corporate boards creates a major and positive impact on shaping firm value and performance. In the previous era governments have globally taken measures for the inclusion of women on corporate boards of firms by making amendments in law, but in real-time the process of inclusion of women on boards of firms is somewhat slow. Many researchers have investigated gender diversity on boards at the firm level. However,

these studies have largely focused on developed markets, and have failed to examine the features of developing markets. It is important to mention here that developing countries reveal the different features of formal, cultural, and ethical environments, which play a significant role in defining the diversity of boards in general, as well as diversity based on gender in particular. Regardless of increasing awareness about the different features of developing markets, there is a clear lack of studies on individual characteristics that affect board gender diversity of developing economies' firms. We are contributing to the literature in following ways: First, the aim of this study was to fill the gap by concentrating on Asian emerging economies (China, Malaysia, India, Pakistan) and examining the role of women on boards in terms of dividend announcement. Indeed, we observed two aspects of women on board, including the role of women in the presence of family owned businesses, and further investigated how these two factors influence the dividend announcement of a firm. We proposed that board gender diversity and family ownership had a negative impact on the dividend announcement of firm. Prior studies explained similar results using samples of Scandinavian firms, where they failed to find any positive association between board gender diversity and firm strategic financial decisions (Randøy et al. 2006; Palmberg et al. 2009; Marinova et al. 2010; Rose 2007; Joshi et al. 2006; Bøhren and Strøm 2007). Moreover, due to free trade agreements between countries and more globalization, we observed that the negative effect of gender diversity and family ownership was reduced for international firms. The results supported our hypothesis, as Asian emerging economies presented the negative impact of board gender diversity and family ownership on dividend announcements of firms. The findings of our research were contradictory to previous studies that were conducted in established economies that suggest that women on boards positively affect the dividend policy of firms (Nasrum 2013; van Essen et al. 2015; Pucheta-Martínez et al. 2016; Byoun et al. 2016; Chen et al. 2017; Al-Rahahleh 2017; Gyapong et al. 2019). However, Robtus and Cox (1991); Rose (2007); Richard et al. (2004); Attig et al. (2016); Sanan (2019); Jonge (2014); and Liu et al. (2013) have explained that it is not easy for women to emerge in firms. Our results are also in line with previous studies such as that of Bhattacharyya (2007), where the author explained that women's role in family firms is only to support the male members of the family. Therefore, females are less likely to be considered in terms of working at the top position of a firm, as it is not possible for them to break the old school boys' network. Second, it recognizes the situations in which emerging economies disallow women on their boards and how family presence on firms' boards affect the relationship between gender diversity on boards and dividend announcement of firms. Lastly, it extends the prior literature based on agency and resource dependency theory, and explains the significance of adding women on corporate boards of organizations.

This study suggests the guidance for policymakers on the issue of board gender diversity. We found a negative effect of board gender diversity and women in family firms on dividend announcement; this could be due to the mass of women on boards. Most of the companies were only following the legislative requirements for the inclusion of women and not for better policymaking. It is truly alarming that most of the companies failed to even appoint a single woman on their boards, and thus are not following legislation, which is dropping significantly. The finding suggests that policymakers should not view the domestic and international firms with one eye. There should be a spillover on board gender diversity from international to domestic, and international firms should be set as an example for domestic firms for the inclusion of women on boards. It might be the best time for Asian emerging economies to take productive actions for balancing the gender on boardrooms and set a minimum mass of women on boards for better and effective decision making. This action will not only increase the trust of shareholders, but also restrict directors in misusing the sources of shareholders. Lastly, this legislation could help women to get the top executive seats in organizations, breaking the old networks.

Notwithstanding the significant results, this study was not without caveats. Firstly, the study was based on a sample of Asian emerging economies (China, Malaysia, India, and Pakistan), which limited the generalizability of our results to other contexts. A general extension of this work is to study this

relationship in other Asian emerging economies. Secondly, we used only one firm-level variable as a moderator—family ownership to determine the relation between board gender diversity and dividend announcement. It would be interesting in future research to examine other factors such as foreign ownership, top-level gender diversity, and processes for the appointment of women on boards including how they moderate the relationship between gender diversity and dividend announcement. It might be possible that the presence of women on boards of international firms is only symbolic, and thus precautions should be taken in result generalization as their capacity of application might be limited due to the different context of Asian emerging economies, where social acceptance of female equality and inclusion is still in the developing phase. It would be interesting to examine the internationalization as impulse of appointing women on boards by intensely studying its positive impact on economic and social outcome. Thirdly, it would be interesting to know how country-level variables (gender level parity, protection of minority shareholders' interests, ethical behavior of firms) moderate the relationship of board gender diversity and dividend announcement of firms. Finally, Pakistan is among the first Asian emerging economies besides India and Malaysia, which has recently in July 2017 passed the resolution for the appointment of women on corporate boards; thus, it would be interesting to know the degree to which gender proportion is fruitful for countries where institutional improvement is quite low, such as in Pakistan. The literature would be assisted from the investigation of this relationship in the post quota period.

**Author Contributions:** A.M. reviewed the relevant literature and wrote the first draft of the article; A.S. contributed to research positioning and manuscript development; M.A. conceived, designed, and performed the experiments; S.A. analyzed the data; and proofread the paper. All authors have read and agreed to the published version of the manuscript.

**Funding:** This research received no external funding.

**Conflicts of Interest:** The authors declare no conflict of interest.

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
