# Peer review of "Board-Gender Diversity, Family Ownership, and Dividend Announcement: Evidence from Asian Emerging Economies"

_jrfm, doi:10.3390/jrfm13040062_

Round 1
Reviewer 1 Report
English language proofing needed. Several sentences and phrases require rewriting. The manuscript is difficult to read and to comprehend. Numerous typos and grammatical errors need to be corrected. Page 7 line 223 has a double full stop. Page 7 line 227 has a full and then after a word beginning with a small letter instead of a capital letter. Page 3 number 93 is unclear. Is it principle or principal? Who is the principal and who is the agent? Arguments for hypotheses 1 and 2 do not provide the logic, literature or empirical evidence regarding the ‘dividend announcement’ in these two hypotheses. Reading the texts before each of these hypotheses says nothing about dividend announcement. If it is difficult to find literature to support these hypotheses regarding dividend announcement, it is best to frame the logic to provide an argument for the hypothesized relations. This is lacking in H1 and H2 arguments. The literature review also looks disjointed. The authors should provide a table summarizing the key findings of previous studies regarding gender diversity and women representation on boards. It is not very clear how the analysis was done. Is this a panel data analysis? Is the data a cross-sectional and longitudinal data, if so, the authors should state the assumption about the error term. How was the error term treated regarding fixed effects or random effects? In addition, the authors claimed that the data was collected from four different countries. How was the analysis done? Combined sample panel? is there the possibility of analysing each country sample separately? Discussion and contributions of the paper could be improved. What is the unique contribution of the paper compared to previous studies? What could be some of the limitations of this study (apart from the symbolic nature of women on boards and the internationalization issue as an option for future studies)?
Author Response
Dear Reviewer,
Thank you for your positive assessment of the potential of our paper. We subsequently address the comments made by you. As you suggest, we have improved our article and added more references to previous studies.
Below, we provide our response to the specific comments that you raised.
Point 1: English language proofing needed. Several sentences and phrases require rewriting. The manuscript is difficult to read and to comprehend. Numerous typos and grammatical errors need to be corrected.
Response: Please accept our sincere thanks for providing very constructive and developmental feedback. We have re-edited our manuscript and have removed the grammatical mistake and this task is done by the language expert.
Point 2: Page 7 line 223 has a double full stop.
Response: Double full stop on page 7 line 223 is edited and now it is on page 11 line 296 after incorporation of changes.
Point 3: Page 7 line 227 has a full and then after a word beginning with a small letter instead of a capital letter.
Response: The mention line has been edited and now it is on page 11 line 301 instead of page 7 line 227.
Point 4: Page 3 number 93 is unclear. Is it principle or principal? Who is the principal and who is the agent?
Response: On page 3 line number 93, we were trying to explain the principal (shareholders) management (agent) relationship but due to typing error it was written as principle instead of principal. The mistake has been removed and now it is on page 4 line number 107 after incorporation of changes.
Point 5t: Arguments for hypotheses 1 and 2 do not provide the logic, literature or empirical evidence regarding the ‘dividend announcement’ in these two hypotheses. Reading the texts before each of these hypotheses says nothing about dividend announcement. If it is difficult to find literature to support these hypotheses regarding dividend announcements, it is best to frame the logic to provide an argument for the hypothesized relations. This is lacking in H1 and H2 arguments.
Response: Thank you for highlighting this point. In the revised draft we have provided the revised hypotheses with respect to gender diversity and dividend policy.
Point 6: The literature review also looks disjointed. The authors should provide a table summarizing the key findings of previous studies regarding gender diversity and women representation on boards.
Response: We have revised the literature with empirical evidence and provided the summary of previous research in the form of a table which is on page 5 line number 135.
Point 7: It is not very clear how the analysis was done. Is this a panel data analysis? Is the data a cross-sectional and longitudinal data, if so, the authors should state the assumption about the error term. How was the error term treated regarding fixed effects or random effects?
Response: We have done the analysis using the Tobit regression model. Data nature is panel and cross sectional. We have normally distributed the error term and 99% outlier removed for the normality of data.
Point 8: In addition, the authors claimed that the data was collected from four different countries. How was the analysis done? Combined sample panel? Is there the possibility of analyzing each country sample separately?
Response: Thank you for highlighting this point. We have run the test on full sample on an aggregate basis rather on specific basis, as in statistics lower the sample size will lead to higher chances of error.
Point 9: Discussion and contributions of the paper could be improved.
Response: We have improved the discussion section on page 16 and have incorporated previous studies.
Point 10: What is the unique contribution of the paper compared to previous studies?
Response: First, the study is unique in the context that it is first to examine that how women on board influence the dividend announcement in the context of Asian emerging economies (China, India, Malaysia, and Pakistan ) collectively by concentrating on firm features that are exclusive to their business framework. Third, it recognizes the situations in which emerging economies disallow the women on their boards and how family presence on firms board effect the relationship between gender diversity on the board and dividend announcement of firms. Lastly, it extends the prior literature based on agency and resource dependency theory and explains the significance of adding women on corporate boards of organizations.
Point 11: What could be some of the limitations of this study (apart from the symbolic nature of women on boards and the internationalization issue as an option for future studies)?
Response: The first limitation is that we have selected a specific number of countries for this study based on the MSCI emerging economies index. We have only used one Industry-level variable which is family ownership, as there are so many other industry-level variables that may affect the relation of board gender diversity and dividend announcement.
We have revised the paper to improve the logic and arguments. We hope that the paper meets your expectations, as it has improved thanks to your recommendations.

Reviewer 2 Report
Dear Authors:
Your paper/manuscript discusses an interesting and timely topic; i.e., board gender diversity. However, and at the risk of being too direct for which I apologize in advance, it is hard for me to follow due to its shortcomings in English, especially for a scholarly publication where very specific words and attention to details are expected in every way. This may not be a fair assessment of the merit of your work but it did distract me significantly.
That being said, I find your research approach to have some merit and would like for you to consider the following feedback points:
Foreign ownership: See: McGuinness, Paul B., João Paulo Vieito, and Mingzhu Wang. "The role of board gender and foreign ownership in the CSR performance of Chinese listed firms." Journal of Corporate Finance 42 (2017): 75-99. You consider 3 hypotheses. More might be preferable, especially in regards to foreign ownership in the firms located in the emerging countries that you selected. Foreign ownership and/or participation in the overall portfolio of any emerging market company, especially coming from Scandinavia, may have an effect on female representation. See: Geely (China) which acquired the Swedish company Volvo Cars and a stake in Volvo Trucks. In regards to Scandinavia, this study on Denmark may be useful to you: Rose, Caspar. "Does female board representation influence firm performance? The Danish evidence." Corporate Governance: An International Review 15, no. 2 (2007): 404-413. Hong-Kong may represent an interesting case based on point #1. Country selection: China, India, Pakistan, and Malaysia must be further explained, especially in regards to other countries of interest, such as Indonesia, Philippines, in the region. City selection: would a more internationally/globally-connected city within an emerging have more females on its board than another company in the same country which may be more land-locked or slightly outside of the urban centers? Industry selection: can you identify some "male-dominated" industries, such as investment banking and Wall Street in the US vs. more "feminine" industries such as HR? Your 2017 Deloitte study has been updated with another one in regards to China (2019): Women in the Boardroom 2019
Deloitte China Center for Corporate Governance. https://www2.deloitte.com/content/dam/Deloitte/cn/Documents/about-deloitte/deloitte-cn-women-in-the-boardroom-2019-en-190627.pdf You might want to look at it and integrate its findings, as needed. Specificity: throughout your paper, when you mention impact, I would recommend you specify (positive or negative). Similarly, when discussing family ownership, are you implying a privately-held company? As you know, plenty of family-owned business are controlled by families. Therefore, some specificity and clarification to explicitly make your points would welcome.
I hope that the above comments are useful and that you will be able to improve your paper/manuscript.
Best wishes,
Anonymous Reviewer
Author Response
Dear Reviewer,
Thank you for your positive assessment of the potential of our paper. We subsequently address the comments made by you. As you suggest, we have improved our article and added more references to previous studies.
Below, we provide our response to the specific comments that you raised.
Point 1: Your paper/manuscript discusses an interesting and timely topic; i.e., board gender diversity. However, and at the risk of being too direct for which I apologize in advance, it is hard for me to follow due to its shortcomings in English, especially for a scholarly publication where very specific words and attention to details are expected in every way.
Response: Please accept our sincere thanks for providing very constructive and developmental feedback. We have re-edited our manuscript and have removed the grammatical mistake and this task is done by the language expert.
Point 2: Foreign ownership: See: McGuinness, Paul B., João Paulo Vieito, and Mingzhu Wang. "The role of board gender and foreign ownership in the CSR performance of Chinese listed firms." Journal of Corporate Finance 42 (2017): 75-99. You consider 3 hypotheses. More might be preferable, especially in regards to foreign ownership in the firms located in the emerging countries that you selected. Foreign ownership and/or participation in the overall portfolio of any emerging market company, especially coming from Scandinavia, may have an effect on female representation. See: Geely (China) which acquired the Swedish company Volvo Cars and a stake in Volvo Trucks. In regards to Scandinavia, this study on Denmark may be useful to you: Rose, Caspar. "Does female board representation influence firm performance? The Danish evidence." Corporate Governance: An International Review 15, no. 2 (2007): 404-413. Hong-Kong may represent an interesting case based on point #1.
Response: Thank you for raising this point. We have treated foreign ownership as a limitation of this study as we have focused solely on family ownership of firms. As per your direction, we have included the results from Scandinavia on page 16 line number 437.
Point 3: Country selection: China, India, Pakistan, and Malaysia must be further explained, especially in regards to other countries of interest, such as Indonesia, Philippines, in the region.
Response: We have selected the sample of Asian emerging economies based on MSCI (Morgan and Stanley Capital International).The main reason behind the selection of this sample was the common culture and norms and restrictions on women free mobility from domestic to public life.
Second, According to World trade organization (2019), these countries are significant players in international markets due to their human resources and rapid growth. Literature suggests that the progress of these countries will affect the growth of developed economies negatively and will represent the world economies by 2030 (Wilson and Purushothaman 2003; Saeed and Sameer 2017). Third, to increase investor confidence on firms, these countries have taken a progressive measure in the field of reporting and governance.
Point 4: City selection: would a more internationally/globally-connected city within an emerging have more females on its board than another company in the same country which may be more land-locked or slightly outside of the urban centers?
Response: Thank you for highlighting an interesting point. We have selected the sample based on a firm registered on a country main stock exchange. City selection would be an interesting topic to study in future. But unfortunately due to the unavailability of data we were not able to corporate this point.
Point 5: Industry selection: can you identify some "male-dominated" industries, such as investment banking and Wall Street in the US vs. more "feminine" industries such as HR?
Response: We have incorporated the male dominant industries on page 10 line number 253.
Point 6: Your 2017 Deloitte study has been updated with another one in regards to China (2019): Women in the Boardroom 2019.Deloitte China Center for Corporate Governance. https://www2.deloitte.com/content/dam/Deloitte/cn/Documents/about-deloitte/deloitte-cn-women-in-the-boardroom-2019-en-190627.pdf
Response: We have incorporated the changes based on the updated report.
Point 7: You might want to look at it and integrate its findings, as needed. Specificity: throughout your paper, when you mention impact, I would recommend you specify (positive or negative).
Response: We have incorporated this point and have revised our manuscript accordingly.
Point 8: Similarly, when discussing family ownership, are you implying a privately-held company? As you know, plenty of family-owned business are controlled by families. Therefore, some specificity and clarification to explicitly make your points would welcome.
Response: We have selected the firm based on ownership concentration if specific person or family is holding 10 % or more shares and holding more board seats, the firm is treated as family firm.
We have revised the paper to improve the logic and arguments. We hope that the paper meets your expectations, as it has improved thanks to your recommendations.
Round 2
Reviewer 1 Report
The version has been much improved than previous versions. I am ok with the revised manuscript. However, the paper still requires proofing.
Author Response
Dear Reviewer,
Thank you for your positive assessment of the potential of our paper. Please accept our sincere thanks for providing very constructive and developmental feedback. We have re-edited our manuscript and have removed the grammatical mistake.
We hope that the paper meets your expectations.
Thank you
Reviewer 2 Report
Dear Author(s):
You diligently made most of the changes that I had suggested. I think that your manuscript is more complete and convincing now, especially from the literature review's perspective.
However, I still see some minor issues that copy-editing will probably rectify (e.g., on p. 2, line 45: "in the light.." on p. 17, line 449: awkward wording "seeing boom in internationalization" or on p. 18, line 454: " in contradict") and others (e.g., the percentage increase from 15% to 16.9% is a 12.66% increase, not 1.9%) that you must correct with another careful review on your own. For example, your discussion and conclusions could be written more clearly and with a more robust structure (maybe with a brief table) on what studies' specific findings your study confirms or contradicts, what your contributions are, and what limitations your study still suffer from.
I hope that these comments are useful.
Best wishes.
Your Reviewer
Author Response
Dear Reviewer,
Thank you for your positive assessment of the potential of our paper. We subsequently address the comments made by you. As you suggest, we have improved our article and added more references to previous studies.
Below, we provide our response to the specific comments that you raised.
Point 1: I still see some minor issues that copy-editing will probably rectify (e.g., on p. 2, line 45: "in the light.." on p. 17, line 449: awkward wording "seeing boom in internationalization" or on p. 18, line 454: " in contradict")
Response: Please accept our sincere thanks for providing very constructive and developmental feedback. We have re-edited our manuscript as per your guideline.
Point 2: the percentage increase from 15% to 16.9% is a 12.66% increase, not 1.9%.
Response: Thank you for pointing out the issue. As per Deloitte Global’s sixth edition of Women in the Boardroom: A Global Perspective reports that women hold just 16.9 percent of board seats globally, a 1.9 percent increase from the report’s last edition published in 2017.
Point 3: your discussion and conclusions could be written more clearly and with a more robust structure (maybe with a brief table) on what studies' specific findings your study confirms or contradicts, what your contributions are, and what limitations your study still suffers from.
Response: We have re-edited the highlighted sections of our manuscript as per your suggestions.